# Low Temperature Delays the Effects of Ischemia in Bergmann Glia and in Cerebellar Tissue Swelling

**DOI:** 10.3390/biomedicines11051363

**Published:** 2023-05-05

**Authors:** Xia Li, Romain Helleringer, Lora L. Martucci, Glenn Dallérac, José-Manuel Cancela, Micaela Galante

**Affiliations:** Institut des Neurosciences Paris-Saclay, CNRS, Université Paris-Saclay, 91400 Saclay, France; xia.li@universite-paris-saclay.fr (X.L.); hellerin@biologie.ens.fr (R.H.); lora.martucci@new.ox.ac.uk (L.L.M.); glenn.dallerac@universite-paris-saclay.fr (G.D.); jose-manuel.cancela@universite-paris-saclay.fr (J.-M.C.)

**Keywords:** Bergmann glia, brain ischemia, swelling, patch clamp, oxygen and glucose deprivation, cerebellum, astrocyte

## Abstract

Cerebral ischemia results in oxygen and glucose deprivation that most commonly occurs after a reduction or interruption in the blood supply to the brain. The consequences of cerebral ischemia are complex and involve the loss of metabolic ATP, excessive K^+^ and glutamate accumulation in the extracellular space, electrolyte imbalance, and brain edema formation. So far, several treatments have been proposed to alleviate ischemic damage, yet few are effective. Here, we focused on the neuroprotective role of lowering the temperature in ischemia mimicked by an episode of oxygen and glucose deprivation (OGD) in mouse cerebellar slices. Our results suggest that lowering the temperature of the extracellular ‘milieu’ delays both the increases in [K^+^]_e_ and tissue swelling, two dreaded consequences of cerebellar ischemia. Moreover, radial glial cells (Bergmann glia) display morphological changes and membrane depolarizations that are markedly impeded by lowering the temperature. Overall, in this model of cerebellar ischemia, hypothermia reduces the deleterious homeostatic changes regulated by Bergmann glia.

## 1. Introduction

The skull represents a fundamental physical barrier protecting the brain against challenges from the external environment. However, its rigidity also imposes strict limits on the capacity of the brain to expand in response to osmotic changes which can have important consequences on neuronal excitability and on circuit activity. In spite of a tight control of brain water homeostasis [1], deleterious distribution of water in the neural tissue can occur following strokes, brain tumors, and neurotrauma, giving rise to the formation of edemas, eventually worsening the clinical situation. One of the few clinical approaches to limit brain edema following ischemia is hypothermia [2,3]. In neonatal ischemic–hypoxic encephalopathy, a disease involving whole-brain ischemia, mild cerebral hypothermia (32–35 °C) is neuroprotective [4,5]. Hypothermia also improves survival and neurological outcome after cardiac arrest and after traumatic brain injury [6].

Studies carried out in rodents have confirmed the efficacy of hypothermia [7], demonstrating beneficial effects including delayed metabolic ATP depletion [8], reduction in excitotoxicity [8,9], and blood–brain barrier preservation [10].

During the acute phase, brain ischemia induces a dramatic imbalance in water homeostasis [1,11] coupled with a gradual cessation of active ionic transport. This leads to a general dysregulation of ionic gradients in cells, mostly characterized by an excess of K^+^ accumulating in the extracellular space associated with an uncontrolled increase in neurotransmitter release [12,13]. Astroglial cells are ideal candidates to counteract the deleterious extracellular K^+^ increase during ischemia, as they express high levels of specific channels such as Kir4.1, Na^+^/K^+^-ATPases, and ion transporters to efficiently clear excess K^+^ [14,15]. Moreover, they are the brain cells that most abundantly express water channels, in the form of the aquaporin AQP4 that contributes to astrocyte swelling during extracellular K^+^ buffering [16,17]. Finally, astrocytes form a network of cells interconnected via gap junctions, sharing ions and water in an extended syncytium, hence favoring their redistribution and eventual expulsion to subpial endfeet and blood vessels.

Bergmann cells are a type of radial glia specific to the cerebellar cortex. They share many features with astrocytes including the prominent expression of Kir4.1 [18,19,20] and AQP4 channels [20,21]. They are also organized as a syncytium of electrically coupled cells [22,23] allowing water and ions to redistribute within the network, thus suggesting an important role in their regulation, especially in pathophysiological conditions such as ischemia. Bergmann glia cell bodies are hosted in the Purkinje cell layer of the cerebellar cortex, where they largely outnumber neurons (up to eight Bergmann cells surrounding one Purkinje soma in rodents [24]). Such high density, coupled with the morphological features of Bergmann cells, forming a radial palisade encompassing the full depth of the molecular layer, implies a tight neuroglial proximity that is further represented by the high coverage of Purkinje neuron synapses by glial membranes [25]. 

In a previous study, we showed that Bergmann glial cells respond to ischemia mimicked by an episode of oxygen and glucose deprivation (OGD) with a prolonged and reversible membrane depolarization, mainly due to the accumulation of K^+^ in the extracellular space [26]. In the present study, we examine the effects of lowering the temperature on this response to OGD using a time-lapse analysis of edema formation and Bergmann cells’ volume regulation together with their membrane depolarization and extracellular K^+^ concentrations. Lowering the temperature yielded a marked decrease in the kinetics of cerebellar edema development and in the Bergmann glia responses to OGD. 

## 2. Materials and Methods

### 2.1. Cerebellar Slice Preparation

All mice were housed on a standard 12 h light/dark cycle with food and water ad libitum. The experimental procedures were in strict accordance with European legislation (2010/63EU Council Directive Decree) and followed Annex IV of the French Decree (1 February 2013) establishing the guidelines for euthanasia. Experimental protocols were approved by the animal welfare body of our institution (Institut des Neurosciences, NeuroPSI). All efforts were made to minimize animal suffering and to reduce the number of animals used in this study. 

Acute cerebellar slices were made from C57Bl/6J mice and Tg(Aldh1l1-EGFP)OFC789Gsat/Mmucd male mice aged 2 to 4 months (for simplicity, we will call them Aldh1l1-EGFP mice). Following deep anesthesia with isoflurane, the mice were decapitated, and the cerebellum was quickly removed in an ice-cold artificial cerebrospinal fluid (ACSF) containing the following (mM): 124 NaCl, 3 KCl, 1.15 KH_2_PO_4_, 1.15 MgSO_4_, 24 NaHCO_3_, 10 Glucose, 2 CaCl_2_ (osmolarity: 300 mOsm and pH 7.35, bubbled with 5% CO_2_, 95% O_2_). Slices were cut at room temperature using a vibrating blade microtome (650 HV, Microm Microtech France, Brignais, France) in ACSF supplemented with an antagonist of NMDARs (APV 50 µM). 

During experiments, the slices were placed in a recording chamber mounted on an upright microscope where cerebellar slices were continuously superfused with ACSF at a rate of 2 mL/min. The temperature of the extracellular solution was maintained at either 31–33 °C or 21–23 °C. To simulate an extracellular ischemic condition, glucose was substituted by sucrose (10 mM) and the O_2_ by N_2_ (95% with 5% CO_2_) in the ACSF (OGD solution).

### 2.2. Tissue Swelling Assay

To analyze the effect of ischemia on tissue swelling, the light transmittance was recorded [27] through a CCD camera (Coolsnap cf, Teledyne Photometrics, Krailling, Germany) and a 40× water-immersion objective mounted on an upright microscope (Axioscope FS, Zeiss, Rueil Malmaison, France). The average intensity of transmitted light was measured in the entire optic field using Metavue software (Molecular Device). The relative change in light transmittance Δ*T*/*T* was calculated as:$$\frac{\Delta T}{T}=\frac{\left(Tn-T0\right)}{T0}\times 100$$
where *T*0 is the mean of the transmitted light before OGD, and *Tn* is the transmitted light acquired at time *n*. The relative variation in *T* was then evaluated as % during the OGD. 

### 2.3. Confocal Microscopy and Image Acquisition

Images were recorded using an upright eclipse FN1 NIKON laser scanning confocal microscope (A1R) equipped with a 488 nm laser and an objective 40× (N.A. 0.8) HCX APO water-immersion objective. The NIKON software NIS Element AR4.50.00 was used for image acquisition. EGFP was excited at 488 nm, and the emission was recorded with a bandpass emission filter from 500 nm to 550 nm. 

The parameters for acquisition of 3-dimensional images were: 512 × 512 pixels with a z-directional interval of 1.5 µm for cell somata and 1024 × 1024 pixel with a z-directional interval of 1 µm for glia limitans. Final voxel size ranged from 0.1 × 0.1 × 1 µm to 0.3 × 0.3 × 1.5 µm. 

### 2.4. Image Processing

Bergmann cell soma and processes were identified by EGFP expression in Aldh1l1-EGFP mice, and image analysis was performed with NIH ImageJ (https://imagej.net/Fiji accessed on 30 October 2018). Images were first smoothed and filtered in order to reduce the image noise. Then, background noise was subtracted with the ‘rolling ball’ algorithm, and a threshold of pixels was fixed. Finally, the volume of the cell soma/process was estimated by the Cavalieri method [28,29,30] consisting in summing up the total area of each stack ‘Si’ multiplied by the z-directional interval ‘*Z*’: $$V=Z\overset{n}{\underset{i=1}{\sum }}S_{i}$$
where ‘*n*’ is the number of stacks.

Then, for each cell, the *V* values were normalized to the mean *V* calculated 6 min before OGD.

### 2.5. Electrophysiology

Whole-cell patch-clamp recordings of Bergmann cells were performed with an Axopatch 200 amplifier. Patch pipettes were pulled from borosilicate glass capillaries and had a resistance of 6–7 MΩ when filled with an intracellular solution containing (mM): K-gluconate 140, MgCl_2_ 1, KCl 4, Hepes 10, EGTA 0.75, Na_2_ATP 4, NaGTP 0.4 (~300 mOsm and pH 7.35). Cells with series resistance deviating more than 20% were excluded. The membrane potential of Bergmann cells was held at −70 mV in voltage clamp mode. Data analysis was performed with Clampfit (Molecular Devices, Wokingham, UK) and Igor (WaveMetrics, Portland, OR, USA) software. The total electrical charge transferred through the membrane during OGD was calculated as the integral of the current recorded during the ischemic protocol (I_OGD_) over its entire duration (30 min). 

### 2.6. Ion-Sensitive Microelectrode Recordings

The ion-sensitive microelectrodes (ISMs) were prepared using a procedure previously reported [26]. Briefly, glass capillaries were silanized with dimethylchlorosilane, dried at 120 °C, and filled with the potassium ionophore I-cocktail B (Merck, Saint-Quentin-Fallavier, France) or the liquid ion exchanger (IE 190, WPI) in the tip. For K^+^-sensitive microelectrodes (K+-ISMs), the rest of the pipette was backfilled with a solution of KCl at 200 mM. Using an ion-sensitive amplifier (ION-01M, npi electronic, Tamm, Germany), the difference in electric potential between the inner and outer side of the ionophore was determined. The K^+^-ISMs were calibrated in the ASCF at different KCl concentrations (4.15 mM, 8 mM, 20 mM, 60 mM, 200 mM). Only ion-sensitive pipettes with stable recordings during the calibration were used for experiments. In order to convert the voltage signal to [K^+^]_e_, we used the Nernst equation.

Changes in the extracellular volume in the molecular layer during OGD were determined by measuring variations in tetramethylammonium ion (TMA^+^) concentration. As cell membranes are impermeable to this cation, changes in TMA^+^ extracellular concentration are inversely proportional to extracellular space volume (V_e_). The pipette preparation was similar to that of K^+^-ISM in which the tip contained the liquid ion exchanger (IE 190 from WPI) and a solution of TMACl (100 mM). TMA^+^-ISMs were calibrated in solutions containing 1, 2, 5, and 10 mM TMACl. For measurements of the [TMA^+^]_e_, slices were perfused continuously with ACSF containing 1 mM TMACl. We then calculated changes in V_e_ using the equation:$$V_{e}\left(\%\right)=\frac{V_{e,n}}{V_{e,\;ctr}}=\frac{\left({\left[TMA^{+}\right]}_{e,\;ctr}\right)}{{\left[TMA^{+}\right]}_{e,\;ischemia}}\times 100$$
where *V_e,ctr_* is the mean *V_e_* measure 4 min before ischemia, and *V_e,n_* is the *V_e_* calculated at time *n*.

For both K^+^-ISM and TMA^+^-ISM, the microelectrodes were placed in the middle of the molecular layer at a depth of ~100 µm.

### 2.7. Data Analysis and Statistics 

Acquisition of electrophysiological data was achieved using Elphy software (G. Sadoc, France). Data analysis was performed offline using Clampfit (Molecular Devices, Wokingham, UK) and Igor (WaveMetrics, Portland, OR, USA) software. All data are presented as mean ± SEM. Statistics were performed with one-way or two-way Anova and with non-parametric tests (Mann–Whitney and Wilcoxon rank test) when samples were too small (n < 10). α level was set at 0.05.

The rise time was considered as the interval between 10% and 90% of the maximal value of the variable studied (ΔT/T, V_e_, I_OGD_, [K^+^]_e_). For the analysis of volume changes in small processes at the glial endfeet, it was hard to precisely detect 10% and 90% of the peak variation; therefore, we measured the time from OGD beginning and 25% of the maximal volume variation (latency). 

## 3. Results

### 3.1. Low Temperature Delays Cerebellar Tissue Swelling during Ischemia

A severe consequence of cerebral ischemia is edema, the abnormal accumulation of fluid that leads to detrimental complications and potential death [31]. To study the effects of the temperature on cerebellar edema, we used a classical OGD protocol consisting in perfusing the slices with an extracellular solution lacking O_2_ and glucose in 2 different thermic conditions: 31–33 °C or 21–23 °C. We focused in particular on the molecular layer presenting a high density of fibers, Purkinje cell dendrites, and Bergman glia processes, in which electrical responses to ischemia have already been described [8,26,32,33]. To determine the effect of the temperature on cerebellar tissue swelling, we simultaneously measured two parameters: the light transmittance (Figure 1A–E) that increases when cells swell, and the extracellular space volume (Figure 1A,F–H) that decreases concomitantly with cell swelling. Figure 1B shows examples of light transmittance changes during ex vivo ischemia (OGD) in the two temperature conditions.

During ischemia, there was a gradual increase in transmitted light of which the kinetics were highly dependent on temperature (n = 7 at 21–23 °C vs. n = 7 at 31–33 °C, *p* = 0.0013, two-way Anova, repeated measures, Figure 1C). This was confirmed by the comparison of ∆T/T rise times: at more physiological temperatures, the rise time was significantly shorter than at lower temperatures (*p* = 0.01, n = 7 and n = 7, respectively, Mann–Whitney test, Figure 1D). The peak value of ∆T/T was, however, similar in the 2 temperature conditions (n = 7 and n = 7, *p* > 0.05, Mann–Whitney test, Figure 1E). Moreover, changes in the extracellular space volume were quantified in parallel to transmitted light recordings using ion-sensitive microelectrodes. To this end, the slice was perfused with low concentrations of tetramethylammonium ions (TMA^+^, 1 mM) that do not permeate cell membranes and were used as an index of changes in extracellular volume (see Section 2). With this approach, a decrease in the extracellular space volume was observed at both temperatures. Similarly to the transmitted light results, extracellular volume variations were significantly delayed at 21–23 °C (Figure 1F, n = 10 at 31–33 °C and n = 8 at 21–23 °C, *p* = 0.0012, two-way Anova, repeated measures) as reflected by a significantly decreased rise time (n = 10 at 31–33 °C and n = 8 at 21–23 °C, *p* = 0.029, Mann–Whitney test, Figure 1G), and the maximal extracellular volume decrease was unchanged (n = 10 at 31–33 °C and n = 8 at 21–23 °C, *p* = 0.019, Mann–Whitney test, Figure 1H). In a subset of experiments, tissue swelling and extracellular volume shrinking were recorded simultaneously and yielded a strong correlation (r^2^: 0.79 ± 0.04 and 0.64 ± 0.1 at low and higher temperatures, respectively, n = 6, Figure 1F inset).

These results indicate that the temperature of the extracellular liquid compartment markedly reduces the kinetics of water exchanges through cerebellar cells during ischemia.

### 3.2. Bergmann Glial Cells Adapt Their Volume to the Extracellular Environment during OGD 

Glial cells are key components of the water homeostatic system of the brain. Their ability to undergo rapid changes in volume allows them to efficiently buffer and respond to modifications in the ionic composition of the extracellular milieu [20,34,35]. In the cerebellum, Bergmann glia radial processes extend throughout the molecular layer, with their fine processes in close vicinity to Purkinje neuron synapses and blood vessels [25]. As these glial cells express water and potassium channels commonly implicated in extracellular space homeostasis [19,21], we sought to specifically analyze the volume change of Bergmann glia during ischemia. To readily visualize Bergmann glia in the molecular layer, experiments were carried out on cerebellar slices from Aldh1l1-EGFP transgenic mice that express the enhanced green fluorescent protein in astrocytes. 

Using real-time confocal imaging, we first focused on the soma of Bergmann glia (Figure 2A). 

Somatic volume in control solution was similar in the 2 temperature conditions (442.7 ± 35.6 µm^3^, n = 7 and 445.8 ± 36.1 µm^3^, n = 10, *p* > 0.05, at low and higher temperatures, respectively), and during OGD the somatic volume decreased gradually in both conditions (Figure 2B,C). This was attributable to OGD and not to other experimental conditions such as photobleaching as we verified that the Bergmann cell soma volume remained stable over a 30 min period of monitoring in the control ACSF (to 99.78 ± 0.88% of the initial size, n = 3, *p* > 0.05, one-way Anova, Figure 2C). Furthermore, reoxygenation and perfusion of the slices with the control solution yielded a partial recovery such that after 18 min the mean volume was 93.6 ± 3.6% of baseline (at 31–33 °C, n = 7). Importantly, here too, performing the OGD under low-temperature conditions resulted in significantly delayed shrinking dynamics (n = 10 at 21–23 °C, n = 7 at 31–33 °C, *p* = 0.025, two-way Anova, repeated measures; Figure 2C) without affecting the maximal volume change (9.3 ± 1.8%, n = 9 at 21–23 °C vs. 13.9 ± 2.0%, n = 7 at 31–33 °C, *p* > 0.05). This effect was also apparent on the volume change latency measured as the time from OGD onset to 25% of peak volume change (n = 9 at 21–23 °C vs. n = 7 at 31–33 °C; *p* < 0.01, Mann–Whitney test, Figure 2D). Together, these results show that ischemia induces a sustained reduction in the Bergmann glia cell body and that the temperature significantly modulates the dynamics of this morphological effect. 

Bergmann glial cells form an extensive network of cells interconnected via gap junctions [22] and further form specialized connections in the form of perivascular and subpial endfeet [36]. Since the latter interface neuropil and fluid compartments, they likely represent an efficient mechanism to exchange an excess of ions and water, as do astrocytes from other brain regions [37,38,39]. We therefore also assessed volume changes in Bergmann glia subpial endfeet forming the glia limitans, readily identifiable in the cerebellar sections of Aldh1l1-EGFP mice (Figure 3A,B). 

In contrast to cell soma, the Bergmann cell endfeet volume increased gradually during OGD (Figure 3A–C). Swelling latency was significantly increased in low (16.0 ± 1.3 min, 21–23 °C, n = 6) compared with higher (10.02 ± 1.1 min, n = 6, Mann–Whitney test, *p* = 0.0203, Figure 3D) temperatures but reached a similar maximal value in the 2 conditions (mean maximal change: 16.6 ± 3.9% at 21–23 °C, n = 6 vs. 17.7 ± 3.3% at 31–33 °C, n = 7, *p* = 0.37, Mann–Whitney test, Figure 3E).

Altogether, our results show that during ischemia the morphological change of the glia limitans is opposite to that of the Bergmann glia cell body, indicating that different parts of this type of glial cell react differently to the ischemic episode. Importantly, lowering the temperature of the extracellular fluid yielded a consistent protective effect on cerebellar tissue by significantly deferring global tissue swelling and the local volume modifications of Bergmann cells.

### 3.3. Temperature Modulates the Kinetics of Bergmann Glia Currents and K^+^ Accumulation during Ischemia

In a previous study, we showed that Bergmann cells depolarize during OGD, with dynamics that are highly correlated to those of [K^+^]_e_ increases [26]. It is thus possible that the effects of temperature on Bergmann glia cell volume are due, at least in part, to a delayed accumulation of extracellular K^+^ during ischemia. To test this hypothesis, we recorded [K^+^]_e_ with potassium-sensitive microelectrodes during OGD. As represented in Figure 4B, at 21–23 °C, [K^+^]_e_ displayed slower kinetics than at more physiological temperatures (n = 17 and n = 7, respectively, *p* < 0.0001, two-way Anova, repeated measures). Moreover, the rise time of the [K^+^]_e_ increases was also significantly dependent on the temperature (n = 17 and n = 7 at low and more physiological temperatures, respectively; *p* = 0.0004, Mann−Whitney, Figure 4C), while the mean peak value of the [K^+^]_e_ increases was similar in both conditions (n = 17 vs. n = 7, *p* = >0.66, Figure 4D). 

Finally, we recorded Bergmann glia membrane currents during ischemia (I_OGD_) using whole-cell patch clamp. I_OGD_ showed slower dynamics when recorded at low temperatures (n = 9 and n = 11 for 21–23 °C and 31–33 °C, respectively, *p* < 0001, two-way Anova), and the rise time was also significantly affected (n = 9 and n = 11 for 21–23 °C and 31–33 °C, respectively, *p* = 0.0011, Figure 4F). Furthermore, the total electric charge passing through the Bergmann glia membrane during ischemia was significantly smaller at low temperatures (n = 8 at 21–23 °C and n = 11 at 31–33 °C; *p* = 0.023, Mann–Whitney test, Figure 4G). 

These data demonstrate that the temperature plays an important role in modulating the kinetics of [K^+^]_e_ accumulation and Bergmann cell depolarization induced by ischemia.

## 4. Discussion

In this study, we examined the effects of low temperature on the physiological consequences of ischemic events. We did so by exposing cerebellar slices from adult mice to episodes of OGD that mimic the early phases of an ischemic attack. Our results show that hypothermia unequivocally defers water and extracellular K^+^ regulation and the Bergmann glia volume changes and their membrane depolarizes during ischemia.

Slower [K^+^]_e_ increases during OGD result in a corresponding temporal shift in Bergmann glia membrane depolarizations, as the latter directly depends on [K^+^]_e_ variations during ischemia [26]. These results are consistent with those reported in the cerebellar cortex at the level of Purkinje cells [8]. In these neurons, lower temperatures produce a twofold increase in the rise time of ischemic depolarizations without changes in the overall amplitude. Moreover, Purkinje neuron ischemic currents induced by OGD are attributable to exacerbated glutamate release [32], which is also delayed by cooling the extracellular solution [8]. 

The mechanisms underlying the multiple effects observed in hypothermic conditions are only partly understood. It is nonetheless widely accepted that hypothermia preserves cerebral metabolism by slowing it down [40]. Notably, lowered temperatures delay the loss of essential metabolic substrates including high-energy phosphate compounds such as ATP. This is important as it protects brain tissues from excitotoxic events triggered by increased lactate production such as acidosis and ATPase dysfunction. This notion, originating from clinical studies, was confirmed in a study performed in cerebellar slices [8]. Our results are in line with these previous findings, since in our experiments the preservation of metabolic ATP at low temperatures likely contributes to maintaining ionic gradients and thus to delaying [K^+^]_e_ increases and Bergmann glia depolarizations. 

The final outcome of the temporal shift produced by a slower metabolism is a reduction in the development of the edema typically associated with ischemia. Indeed, as illustrated in Figure 5A, global tissue swelling and shrinking of the extracellular space volume observed during OGD are also shifted in time with similar latencies (~10 min). Our experiments also suggest that water entry into cellular elements of the molecular layer could explain both the tissue swelling and the reduction in extracellular space that we observed. This result agrees with that observed in rodents, where mild hypothermia reduced the size of ischemic infarct elicited by in vivo global ischemia [41,42].

With respect to Bergmann glia, we interestingly observed that distinct cellular compartments respond differently to ischemia. Namely, Bergmann cell bodies shrink while their subpial endfeet (glia limitans) swell (Figure 5B). A possible explanation for this difference could be the differential expression of ion channels and transporters implicated in water balance in specific compartments of Bergmann glia. The glial water channel AQP4 has been identified as an important factor that controls water balance in the brain [1,43], and Bergmann glia express AQP4 predominantly in perivascular and subpial endfeet [20,21,44]. In these compartments, there are also high densities of Kir4.1 K^+^ channels [19,20] that are important contributors to K^+^ homeostasis. The co-localization of Kir4.1 and AQP4 at the interface with liquid compartments has been observed in astrocytes of other brain regions, and it has been suggested that these channels act in synergy to regulate brain osmolarity [14,37]. The polarized distribution of AQP4 and Kir4.1 channels in Bergmann glia and the poor expression of these channels in the somatic region [20] may thus explain the different reactions of the cell body and subpial endfeet to volume changes. Differences between the volume regulation of soma and processes have also been reported in spinal cord [45], hippocampal [20], and cortical [46] astrocytes. However, Bergmann cells are, to our knowledge, the only known astroglial type exhibiting simultaneous swelling of the processes and somatic shrinkage. We believe that this peculiar characteristic may stem from the unique morphology of the cerebellar cortex. Indeed, a single Purkinje cell body is tightly associated with, on average, eight Bergmann cell soma, which are in an ideal position to buffer changes in ionic concentrations during pathological events. It is thus tempting to speculate that the potent extrusion of K^+^ from Purkinje cells hyper-excited during ischemic events leads to the exit of water from Bergmann soma and thus to their shrinking (Figure 5B).

In conclusion, our results show that the main effect of hypothermia is a temporal shift in the acute events characterizing an ischemic episode. Nevertheless, a remaining open question is whether other mechanisms are at play in the neuroprotection that hypothermia provides against ischemic insults. More studies are thus required to complete our understanding of the pathological events triggered by a sudden decrease in blood flow in the brain and of the beneficial effects of hypothermia in order to identify new therapeutic strategies or refine current approaches to counteract this dramatic event. 

## Figures and Tables

**Figure 1 biomedicines-11-01363-f001:**
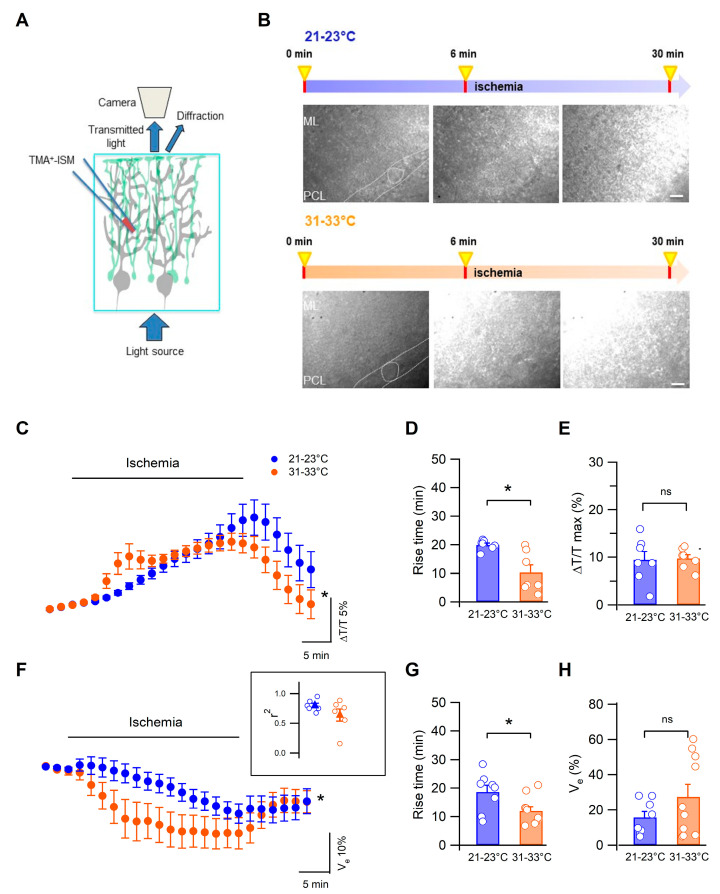
Measurements of light transmitted and extracellular space volume in the molecular layer. (**A**) Scheme illustrating Bergmann glia syncytium (green) and Purkinje neurons (gray) in a parasagittal plane. The TMA^+^-ISM and the principle of transmittance detection are shown. (**B**) Images of 2 cerebellar slices acquired before (0 min), after 6 min, and after 30 min of the oxygen and glucose deprivation protocol. At the top, the slice was perfused at T = 22 °C, while at the bottom the temperature in the ACSF was 32 °C. Dotted lines indicate Purkinje neurons (calibration bar: 10 µm). The visual field was always centered in the molecular layer (ML) close to the Purkinje cell layer (PCL). Notice the increase in light transmittance after 6 min of ischemia at 32 °C. (**C**) Relative transmitted light intensity (ΔT/T) before, during, and after ischemia in the 2 temperature conditions (blue: 21–23 °C, n = 7; orange: 31–33 °C, n = 7, *p* < 0.01). (**D**) ΔT/T rise times in the 2 conditions (*p* = 0.01). (**E**) ΔT/T peak values were similar in both temperature conditions (*p* > 0.05). (**F**) Pooled traces of the time-dependent extracellular volume space changes during ischemia at 21–23 °C (n = 8) or at 31–33 °C (n = 10, two-way Anova, *p* = 0.0012). Inset: correlation analysis between ΔT/T and V_e_ in simultaneous recordings (n = 6). Changes in extracellular space volume show different rise times (**G**) with similar peak values (**H**). In (**C**,**F**), each circle represents the average of 2 min recordings. * indicates *p* < 0.05 and n.s. refers to *p* > 0.05.

**Figure 2 biomedicines-11-01363-f002:**
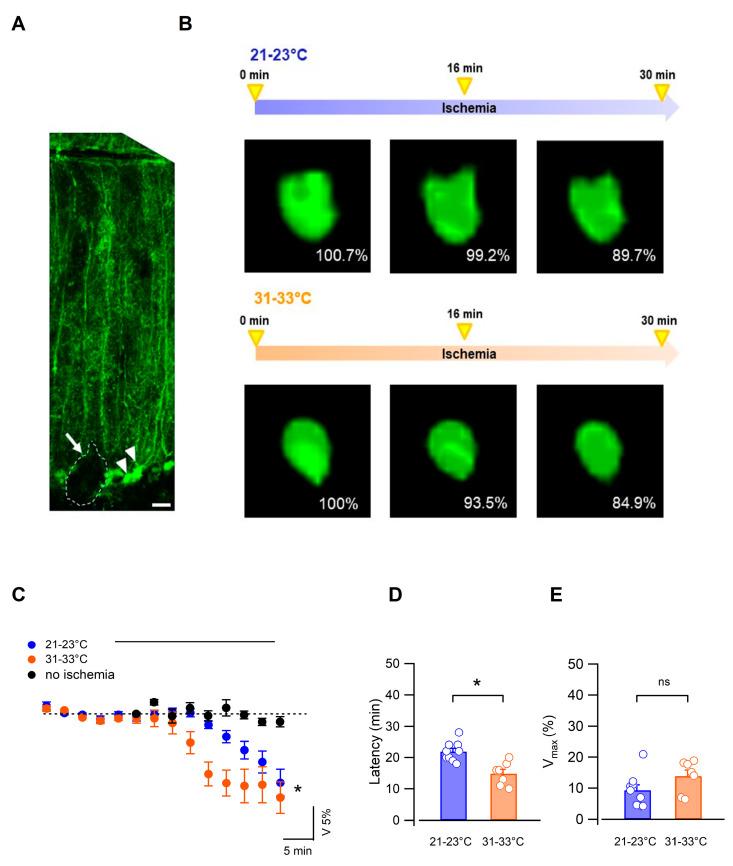
Ischemia induced a temperature-dependent decrease in the Bergmann glia soma volume. (**A**) Confocal image of the Purkinje cell layer and molecular layer of an Aldh1l1-EGFP mouse. Small-size Bergmann glia cell bodies (in green, arrowheads) are located near the soma of a Purkinje cell (dotted line), and their radial processes cover the molecular layer to terminate in flat endfeet at the pial surface (scale bar = 20 μm). (**B**) Images of 2 Bergmann glia cell bodies before and during OGD at 21–23 °C and 31–33 °C. The volume change is indicated as the percentage of the soma volume before OGD protocol. (**C**) Time-dependent changes in soma volumes were quantified using 3D confocal morphometry analysis in each individual cell every 3 min. Ischemia significantly decreased the cell body volume in both temperature conditions but with different temporal evolutions (21–23 °C, n = 10; 31–33 °C, n = 7, *p* = 0.025, two-way Anova). Black circles indicate changes in cell body volume during control experiments in the absence of ischemia (31–33 °C, n = 3). (**D**) The latency of volume change is significantly higher at low temperatures (n = 9 at 21–23 °C vs. n = 7 at 31–33 °C; *p* < 0.01, Mann–Whitney test), while maximal volume changes are not different in the 2 experimental groups (**E**). * indicates *p* < 0.05 and n.s. *p* > 0.05.

**Figure 3 biomedicines-11-01363-f003:**
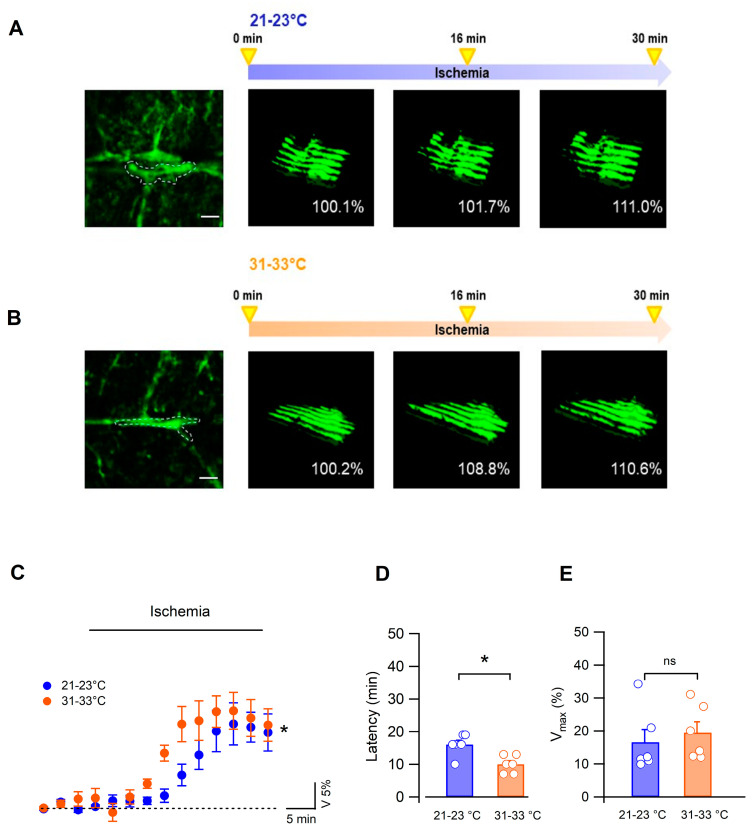
Ischemia induced a swelling of Bergmann glia endfeet. (**A**,**B**) Left: confocal images of Bergmann cell endfeet at the pial surface of the cerebellum in Aldh1l1-EGFP mice (scale bar = 5 µm). Right: Z stacks used to create the 3D reconstruction of the glia limitans in the dotted region of the images at the left. The glia limitans volume was measured before (0 min) and during ischemia in the 2 temperature conditions (scale bar: 5 µm). (**C**) Dynamic changes in Bergmann glia endfeet volume during ischemia are modulated by the temperature of the extracellular solution (n = 6 at low and n = 7 at higher temperatures, two-way Anova, *p* = 0.0443). (**D**) Mean and individual values of ischemia latencies measured at 21–23 °C (n = 6) and 31–33 °C (n = 6, *p* = 0.0203, Mann–Whitney test). (**E**) Maximal values of endfeet volume changes during ischemia are not affected by the temperature (n = 6, *p* = 0.37). * indicates *p* < 0.05 and n.s. *p* > 0.05.

**Figure 4 biomedicines-11-01363-f004:**
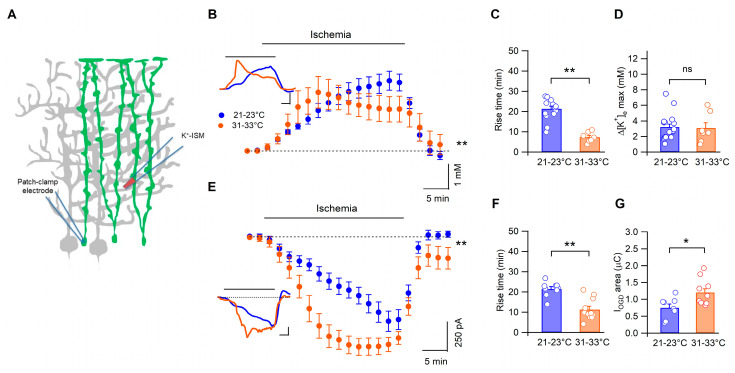
Lowering the temperature of the external milieu delays the kinetics of extracellular K^+^ concentrations and membrane currents during ischemia. (**A**) Scheme indicating the position of both the patch-clamp pipette (Bergmann cell) and the K^+^-sensitive microelectrode. (**B**) Average time-course of [K^+^]_e_ at 21 –23 °C (n = 17) and at 31 –33 °C (n = 7, *p* < 0.0001, two-way Anova). Each point represents the average of 2 min recordings. Inset: two examples of [K^+^]_e_ recordings at low and more physiological temperatures. (**C**) Rise times of [K^+^]_e_ variations are significantly higher at room temperature (n = 17 and n = 7, *p* < 0.001), while (**D**) the peak values were similar in the 2 temperature conditions (*p* > 0.5). (**E**) Mean I_OGD_ recorded in Bergmann glia during ischemia at low (n = 9) and more physiological (n = 11) temperatures (*p* < 0.0001, two-way Anova). Each point represents the average current calculated into 2 min periods. Inset: whole-cell patch-clamp recordings of 2 Bergmann cells during ischemia (holding potential = −70 mV). (**F**) Comparison of I_OGD_ rise times in the 2 experimental groups (n = 9 and n = 11 for 21–23 °C and 31–33 °C, respectively, *p* < 0.001). (**G**) Pooled data of I_OGD_ area revealed that the electrical charge passing through the membrane is significantly reduced by lowering the temperature of the external solution (n = 8 and n = 11 for 21–23 °C and 31–33 °C, respectively, *p* < 0.05, Mann–Whitney test). * indicates *p* < 0.05; n.s. *p* > 0.05 and ** *p* < 0.001.

**Figure 5 biomedicines-11-01363-f005:**
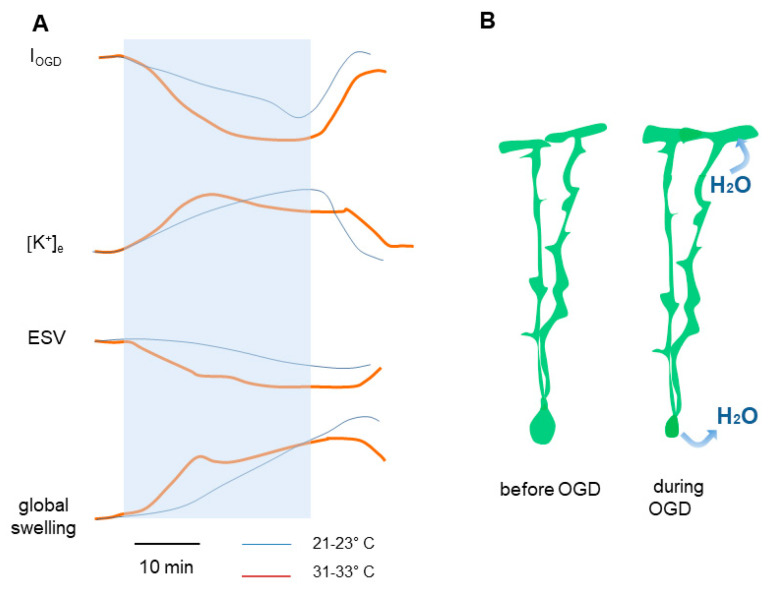
Schematic illustration of temperature effects on parameters measured during cerebellar ischemia. (**A**) Disruption in ion and water homeostasis induces increases in [K^+^]_e_ and Bergmann glia depolarization. In parallel, changes in ion balance in the extracellular space induces a gradual, global tissue swelling and shrinking of the extracellular space volume (ESV). The light blue shadow indicates the OGD. (**B**) Illustration of the volume changes in Bergmann glia during OGD.

## Data Availability

Raw data are available from the corresponding author upon request.

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
