# Peer review of "Low Temperature Delays the Effects of Ischemia in Bergmann Glia and in Cerebellar Tissue Swelling"

_biomedicines, 2023, doi:10.3390/biomedicines11051363_

Round 1

Reviewer 1 Report

This is a solid paper on the effect of hypothermia on cerebellar ischemia. The authors use mouse cerebellar slices and combine electrophysiological and morphological methods elegantly. They focus on radial Glial cells (Bergmann glia) and show that hypothermia can reduce the effects of ischemia on Bergman glia regulation of the extracellular homoestasis. Methods are sophisticated and clear, results are well and clearly presented, discussion is comprehensive.

Specific comments:
The main question, of course, is whether hypothermia can affect damage and/or recovery after neurological injury. This is a relevant and interesting topic, although the answer in general is well known: cooling is beneficial. Thus, the topic is not very „original“. This paper adds the specific behaviour of the Bergmann glia under ischemic and hypothermic conditions, thus adding another piece to a mosaic. The paper is very well written, clear and easy to read, the conclusions are consistent with the evidence and arguments presented and the main question is well addressed.

Author Response

We thank the referee for her/his positive feedback. We agree with the referee that the beneficial effects of cooling the brain are already known and our results confirm this. Our aim here was to focus on the reaction of Bergmann glia during edema formation and define whether hypothermia modulates this response. We believe that our findings provide new insights on the ‘behavior’ of Bergmann glia in response to drastic changes in ion and water contents distinctive of early phases of ischemia. We hope that our study, adding ‘another piece to a mosaic’ will help to understand the role played by astrocytes in the formation of cerebellar edema and the mechanisms by which hypothermia confers neuroprotection in the brain.

Reviewer 2 Report

The authors of the manuscript “Low temperature delays the effects of ischemia in Bergmann 

glia and in cerebellar tissue swelling” designed a set of interesting experiments to address the role of the temperature in regulating fluid osmosis during an experiment model of ischemia, namely OGD, and the function played by Bergmann glia in this process.

Overall, the experimental design is well conceived. Two main points raise doubts: 

1)    Figure 1B: lower and upper panel: The background of the visual field chosen to represent time 0 differs between the two controls. How the authors explain such a difference? Such an image does not seem to represent the value plotted in the corresponding graph.

2)    Figure 2. The variation in volume of the radial processes of the Bergmann glia was not investigated. Did other evidences show no role of these in regulating osmosis during OGD? What’s the role of the other glial cells? Did the authors collect data for microglia, astrocytes and oligodendrocytes? 

Author Response

We thank the referee for reviewing our manuscript on the effects of hypothermia on cerebellar ischemia. A point-by-point response detailing how we have dealt with the points raised by the reviewer is provided below.

1) Referee is right. There was an error in importing the images at low temperatures and we now have inserted the good ones from the original data (see Figure 1B in the revised MS). However, it is important to clarify that even though the background light intensity was slightly different among experiments, during the data analysis we have calculated the variation of the transmitted light T relative to the background which normalizes the measures and reduces variability among experiments. This is explained in Materials and Methods section (lines 98-103) of the MS.

2) In this study, we focused on Bergmann glia end-feet because these contain high densities of AQP4, the most abundant water channel in the brain. It is noteworthy to underline that AQP4 expression is not homogeneous at the astrocytic cell membrane as it is highly enriched in astrocytic perivascular end-feet and glia limitans (Papadopoulos et al., 2007, Nicchia et al., 2008). Therefore, these astrocytic regions, at the interface between the brain and the associated liquid compartments, are strategic for water homeostasis.

Nevertheless, we cannot exclude that Bergmann cell radial processes may also play a role in osmotic regulation during ischemia. During this study, we attempted to quantify the volume of the radial processes of Bergmann glia but the signal-to-noise ratio during confocal microscopy acquisition was too low to obtain stable and consistent recordings.

In contrast, very little information is available in the literature concerning the impact of ischemia on the volume of oligodendrocytes and microglia and on the contribution of these cells to water homeostasis. In fact, even though multiple, sometimes bivalent roles have been described for microglia and oligodendrocytes in ischemic conditions (for review: Hernández et al. 2021, Cells) a clear implication of these cells in water balance has not been reported. Moreover, importantly, Bergmann glial cells are the most abundant astrocyte in the Purkinje cell-molecular layers (Hoogland & DeZeeuw, 2015) whereas both microglia (Stowell et al. 2015) and oligodendrocytes precursors (NG2+-cells, Lin et al., 2005, Neuron) are sparse and present at very low density so that we do not expect a decisive role of these cells in osmotic regulation during ischemia. Still, future studies could examine the involvement of these glial cells, which may provide a more comprehensive understanding of the cellular mechanisms at play.

Round 2

Reviewer 2 Report

No further comments for the authors. My issues were addressed satisfactorily.